# Characterizing neural representation of cognitively-inspired RL agents during an evidence accumulation task

## Abstract

Evidence accumulation is thought to be fundamental for decision-making in humans and other mammals. It has been extensively studied in neuroscience and cognitive science with the goal of explaining how sensory information is sequentially sampled until sufficient evidence has accumulated to favor one decision over others. Neuroscience studies suggest that the hippocampus encodes a low-dimensional ordered representation of evidence through sequential neural activity. Cognitive modelers have proposed a mechanism by which such sequential activity could emerge through the modulation of recurrent weights with a change in the amount of evidence. This gives rise to neurons tuned to a specific magnitude of evidence which resemble neurons recorded in the hippocampus. Here we integrated a cognitive science model inside a Reinforcement Learning (RL) agent and trained the agent to perform a simple evidence accumulation tasks inspired by the behavioral experiments on animals. We compared the agent's performance with the performance of agents equipped with GRUs and RNNs. We found that the agent based on a cognitive model was able to learn faster and generalize better while having significantly fewer parameters. We also compared the emergent neural activity across agents and found that in some cases, GRU-based agents developed similar neural representations to agents based on a cognitive model. This study illustrates how integrating cognitive models and artificial neural networks can lead to brain-like neural representations that can improve learning.

## 1 Introduction

Converging evidence from cognitive science and neuroscience suggests that the brain represents physical and abstract variables in a structured form, as mental or cognitive maps. These maps are thought to play an essential role in learning and reasoning (Tolman, 1948; Ekstrom & Ranganath, 2018; Behrens et al., 2018). Cognitive maps are characterized by neurons that activate sequentially as a function of the magnitude of the variable they encode. For instance, neurons called place cells activate sequentially as a function of spatial distance from some landmark (Moser et al., 2015; Muller, 1996; Sheehan et al., 2021). Similarly, time cells activate sequentially as a function of elapsed time from some event (Pastalkova et al., 2008; MacDonald et al., 2011; Cruzado et al., 2020; Salz et al., 2016).

Similar sequential activity has also been observed for sound frequency (Aronov et al., 2017), probability (Knudsen & Wallis, 2021) and accumulated evidence (Nieh et al., 2021; Morcos & Harvey, 2016b). For example, in the "accumulating towers task" Nieh et al. (2021) trained mice to move along a virtual track and observe objects (towers) on the left- and right-hand sides. When mice arrived at the end of the track, to receive a reward they had to turn left or right, depending on which side had more towers. The difference in the number of towers here corresponds to the amount of evidence for turning left *vs.* turning right. Nieh et al. (2021) recorded activity of hundreds of individual neurons from mice hippocampus, part of the brain commonly thought to play a key role in navigation in physical and abstract spaces (Bures et al., 1997; Eichenbaum, 2014; Moser et al., 2015). The results indicated the existence of cells tuned to a particular difference in the number of towers, such that a population of neurons tiles the entire *evidence* axis (Nieh et al., 2021) (see also Morcos &

Harvey (2016b)). This provides valuable insight into how abstract variables are represented in the brain.

Cognitive scientists have developed elaborate models of evidence accumulation to explain the response time in a variety of behavioral tasks (Laming, 1968; Link, 1975; Ratcliff, 1978). These models hypothesize that the brain contains an internal variable that represents the progress towards the decision. A neural-level cognitive model proposed that the brain could implement this process using a framework based on the Laplace transform (Howard et al., 2018). The Laplace framework gives rise to map-like representations and it has been successful in describing the emergence of sequentially activated time cells (Shankar & Howard, 2012) and place cells (Howard et al., 2014; Howard & Hasselmo, 2020).

Artificial neural networks (ANNs) are commonly thought to have a distributed representation that does not have a map-like structure. While ANNs excel in many domains, they still struggle at many tasks that humans find relatively simple. Unlike humans, ANNs typically require a large number of training examples and fail to generalize to examples that are outside the training distribution (Bengio, 2017; LeVine, 2017; Marcus, 2020). Using cognitive models informed by neural data as an inductive bias for ANNs is an important direction that can help not only advance the current AI systems but also improve our understanding of cognitive mechanisms in the brain.

Here we integrate the Laplace framework into reinforcement learning (RL) agents. The Laplace framework is based on recurrent neurons with analytically computed weights. We use the Laplace domain to generate a map-like representation of the amount of evidence. This representation is then fed into a trainable RL module based on the A2C architecture (Mnih et al., 2016). We compare map-based agents to standard RL agents that use simple recurrent neural networks (RNNs) and Gated Recurrent Units (GRUs) (Chung et al., 2014) in terms of performance and similarity of the neural activity to neural activity recorded in the brain.

Contributions of this work are as follows:

- We integrated a cognitive model for evidence accumulation based on the Laplace transform into an RL agent.

- We showed that symbolic operations in the Laplace domain give rise to individual neurons that are tuned to the magnitude of the evidence, just like neurons in neuroscience studies (Nieh et al., 2021; Morcos & Harvey, 2016a).

- We found that agents based on the Laplace framework learn faster and generalize better than agents based on commonly used RNNs. This indicates that RL agents were able to efficiently use the brain-like sequential representation of evidence.

- We found that GRUs performed much better than RNNs, suggesting that gating plays an important role in constructing a neural representation of time-varying latent variables. This is consistent with the cognitive modeling work, which uses gating to convert a representation of elapsed time into a representation of accumulated evidence.

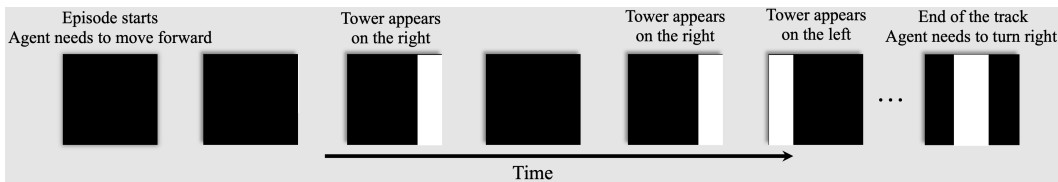

Figure 1: Schematic of the accumulating towers environment. In this simple example, two towers appeared on the right, and one tower appeared on the left, so the agent has to turn right once it reaches the end of the track. Each tower is encoded with a single pixel value.

## 2 METHODS

### 2.1 ENVIRONMENTS

Inspired by the neuroscience studies from Nieh et al. (2021); Morcos & Harvey (2016b), we designed a simple version of the accumulating towers task. To evaluate robustness and generality of the proposed approach, aside from default version of the task, we also designed two other tasks, namely range count task and exact count task. The environments are provided as a part of the supplementary material, and they will be made publicly available and made open source together with the entire code used to implement the artificial agents and generate all of the results in the manuscript.

#### 2.1.1 ACCUMULATING TOWERS TASK

Agents had to navigate down a virtual track composed of only three inputs: left, right and middle. In each episode, agents start from the beginning of the virtual track and observe towers (represented by the input value changing from 0 to 1) on each side of the environment (Fig. 1). Agents had four available actions: left, right, forward, and backward. Positions of towers were decided randomly in each episode. Similar to the neuroscience studies, the maximum number of towers on one side was 14. Once the agent arrived at the end of the track, the middle input changed from zero to one signifying that it hit the wall. In order to receive the reward, the agent had to turn left or right, depending on which side had more towers. We set the magnitude of the reward to 10, penalize the agents for hitting the wall at the end of the track or going backward with a -1 negative reward, and penalize the agent for hitting the side walls with a -0.1 negative reward.

#### 2.1.2 RANGE COUNT TASK

In this task, agents observed the same environment as in the accumulating towers task, but in order to obtain a reward, agents had to turn left only if the number of towers on the left-hand side was larger than five and right in every other case (regardless of the number of towers on the right-hand side).

#### 2.1.3 EXACT COUNT TASK

In the counting task, agents again observed the same environment as in the accumulating towers task, but the maximum number of towers on each side was 5 instead of 14, and agents had to turn left only if the number of towers on the left-hand side was exactly 5 and right in every other case.

### 2.2 RNN IMPLEMENTATION OF EVIDENCE ACCUMULATION USING THE LAPLACE DOMAIN

Evidence accumulation implemented through the Laplace domain is a key component of the cognitively-inspired RL agent (Fig. 2). We will first describe the Laplace framework for functions of time and then convert a representation of time into a representation of evidence and show how it can be implemented as an RNN.

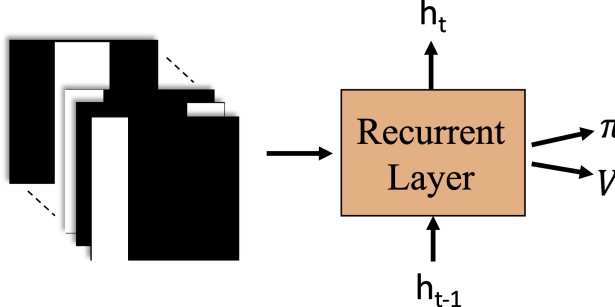

Figure 2: The agent architecture. We compare simple RNN, GRU and Laplace-based RNN described here.

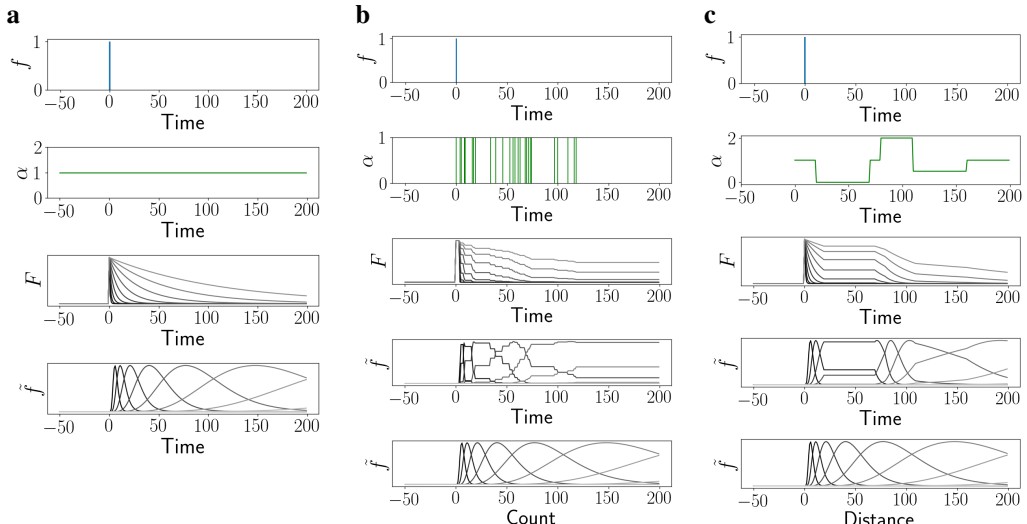

Figure 3: Example of the Laplace and inverse Laplace transform with and without modulatory input. (a) In the absence of modulatory input ($\alpha = 1$) the impulse response of the Laplace transform decays exponentially with decay rate $s$. The impulse response of the inverse Laplace transform has a unimodal shape. Note that if time $t$ was shown on the log-scale, the unimodal curves would be equally wide and equidistant. (b, c) $\alpha$ modulates the decay rate of $F$ and it is proportional to the change in the count (b) or distance (c). This makes units in $\tilde{f}$ develop unimodal basis functions that are tuned to count or distance rather than to time and peak at $n^*$ or $x^*$ respectively.

### 2.2.1 LAPLACE FRAMEWORK

We define the Laplace transform of function $f(t)$ from $-\infty$ to the present $t$:

$$F(s;t) = \int_0^t e^{-s(t-t')} f(t')dt'. \tag{1}$$

We restrict variable $s$ to real positive values. [1]

The above equation can be expressed in a differential form where $s$ appears as a rate constant of a leaky integrator:

$$\frac{dF(s;t)}{dt} = -sF(s;t) + f(t). \tag{2}$$

Fig. 3a shows the impulse response of the above equation for several values of $s$.

To convert a representation of time into a representation of numerosity $n(t)$ (how many times some input was observed) extend Eq. 2 and modulate $s$ with a rate of change $\alpha$ expressed as a time derivative of numerosity ($\alpha = dn/dt$):

$$\frac{dF(s;t)}{dt} = \frac{dn}{dt}\left(-sF(s;t) + f(t)\right). \tag{3}$$

By reorganizing terms in the above equation and applying the chain rule we can rewrite the equation as a function of $n$, instead of $t$ (Fig. 3b):

$$\frac{dF(s;n)}{dn} = -sF(s;n) + f(n), \tag{4}$$

and we set $f(n) = \delta(0)$.

Note that the same approach can convert a function of time into a function of any variable, which derivative can be learned from the environment. For instance, if $\alpha = dx/dt$ is velocity, then the network would represent traveled distance $x$ (Fig. 3c).

---

[1]The Laplace transform defines $s$ as a complex variable. This choice would result in exponentially growing and oscillatory neural activity, causing numerical instabilities when computing the inverse Laplace transform.

Inverting the Laplace transform reconstructs the input as a function of the internal variable $n^*$, which corresponds to $n$. The inverse, which we denote as $\tilde{f}(n^*;t)$ can be computed using the Post inversion formula (Post, 1930):

$$\tilde{f}(n^*;n) = \mathbf{L}_k^{-1} F(s;n) = \frac{(-1)^k}{k!} s^{k+1} \frac{d^k}{ds^k} F(s;n), \tag{5}$$

where $n^* := k/s$ and $k \to \infty$. As we show below, the reconstruction gives rise to units tuned to a particular $n$. By solving $\partial \tilde{f}_{n^*;n}/\partial n = 0$ we see that $\tilde{f}(n^*;n)$ peaks at $n^* = n$. For $s$ being a continuous variable and $k \to \infty$, the width of the peak is infinitesimally small, providing a perfect reconstruction of the observed quantity.

### 2.2.2 DISCRETE IMPLEMENTATION

For a neural network implementation, we discretize the Laplace and inverse Laplace transform for both $s$ and $t$. To select values $s$ in an informed way, we compute the impulse response of $\tilde{f}_{n^*;n}$:

$$\tilde{f}_{n^*;n} = \frac{1}{u(t)} \frac{k^{k+1}}{k!} \left(\frac{u(t)}{n^*}\right)^{k+1} e^{-k\frac{u(t)}{n^*}}, \tag{6}$$

where $u = \sum_{i=0}^{t} \alpha(t_i)$. When $s$ is discrete and $k$ is finite, $\tilde{f}_{n^*;n}$ is a set of unimodal basis functions (when $s$ is continuous and $k \to \infty$, those unimodal basis functions turn into delta functions with spacing $\to 0$). The coefficient of variation of $\tilde{f}_{n^*;n}$ is independent of $n^*$ and $n$: $c = 1/\sqrt{k+1}$. This implies that the width of the unimodal basis functions increases linearly with their peak time. When observed as a function of $\log(n)$, the width of the unimodal basis functions is constant. This property of the Post inversion formula is relevant for modeling human perception due to the Weber-Fechner law (Fechner, 1860/1912; Portugal & Svaiter, 2011). This law states that the relationship between the perceived magnitude of the stimulus and its true magnitude is logarithmic, motivating the use of logarithmic units such as decibel and candela. To ensure equidistant spacing of unimodal basis functions along the log-axis we space $n^*$ logarithmically. This results in dramatic conservation of resources, especially when representing large quantities since the number of units in $\tilde{f}_{n^*;n}$ grows as a function of $log(n)$ rather than $n$. Note that fixing the values of $n^*$ and choosing $k$ also fixes values of $s$ since $s = k/n^*$.

We now write a discrete-time approximation of Eq. equation 3 as an RNN with a diagonal connectivity matrix and a linear activation function:

$$F_{s;t} = W F_{s;t-1} + f_t, \tag{7}$$

where $W = \text{diag}(e^{-\alpha(t)s\Delta t})$. A discrete approximation of the inverse Laplace transform, $\tilde{f}_{n^*;t}$, can be implemented by multiplying $F_{s;t}$ with a derivative matrix $\mathbf{L}_k^{-1}$ computed for some finite value of $k$.

### 2.2.3 SUBTRACTION OF FUNCTIONS USING THE LAPLACE DOMAIN

In the accumulating towers task, Eq. 4 can enable the agent to learn to represent the number of towers on each side. However, the latent variable that should determine the agent's decision is not the number of towers on each side but the difference between those numbers (the agent needs to turn towards the side which had more towers). This is a non-trivial problem since the number of towers is not represented as a scalar but as a function over $n$. Fortuitously, the Laplace domain enables access to a number of useful operations, including subtraction of two functions (Howard et al., 2015). To show this, let us define $f(a)$ and $g(a)$ as functions representing two distributions of possible values for the number $a$ in the range 0 to $a_{max}$. Outside this range, the functions are assumed to vanish. We define the operation of subtraction of these two distributions $[f - g](a)$ to be the cross-correlation of the two functions:

$$[f - g](a) \equiv \int_0^\infty f(x')g(a + x')dx'. \tag{8}$$

To illustrate that the above operation results in subtraction of two functions, consider a simple case where each of the functions is a delta function: $f = \delta(a_1)$ and $g = \delta(a_2)$. Then $[f - g]$ is a delta function at $a_1 - a_2$. To implement cross-correlation in the Laplace domain we can turn Eq. 8

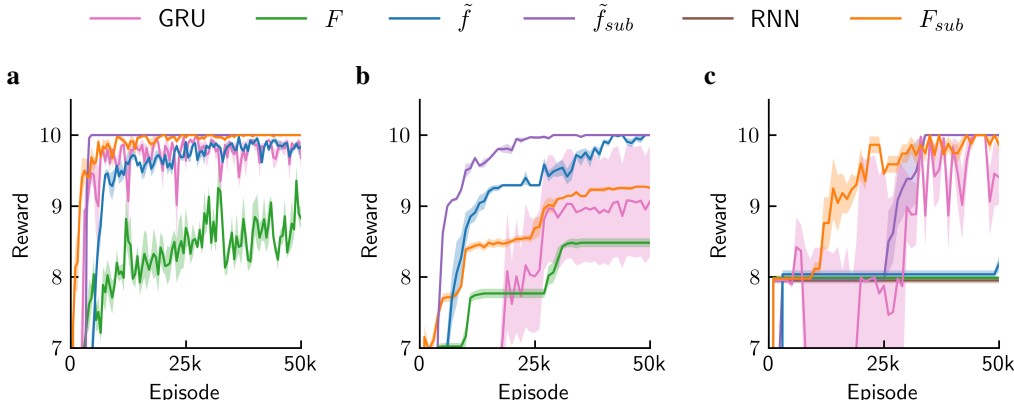

Figure 4: Agent performance on (a) accumulating towers task, (b) range count task, and (c) exact count task. In each task proposed architectures (either $\tilde{f}_{sub}$ or $F_{sub}$) learned the task faster than GRU despite having almost three orders of magnitude fewer parameters.

into convolution by reflecting $g(a)$ around $a_{max}$: $g_r = g(a_{max} - a)$. Point-wise product of the Laplace transforms of two functions, $f(a)$ and $g(a)$, corresponds to their convolution in the time domain. Point-wise multiplication of the Laplace transform of $f(a)$ and $g_r(a)$ corresponds to cross-correlation of $f(a)$ and $g(a)$ in the time domain, which is equivalent to their subtraction $[f - g](a)$. Note that for subtraction we need to consider both positive and negative values. Since we only use positive values of $s$, we are not able to directly represent the negative axis. To work around this, we compute both $[f - g](a)$ and $[g - f](a)$.

## 2.3 AGENT ARCHITECTURE

The agents received three inputs from the environment that were fed into the recurrent layer (Fig. 2). The recurrent layer was either RNN, GRU or an RNN based on the Laplace framework as described above. When the Laplace framework was used we had three independent $\tilde{f}$ modules. Each module had 20 $n^*$ values spaced logarithmically from 5 to 100. The value of parameter $k$ was set to 8. Parameter $k$ controls the sharpness of the unimodal basis functions and this value was chosen to ensure no gaps between them. Note also that the input into $\tilde{f}$ was delivered to $\alpha$, which controls recurrent weights of the population of units $F$. This is a conceptual difference in comparison to other RNNs, where recurrent weights are tuned separately for each unit. The strength of the proposed approach is that the population of neurons encodes a function over a latent variable that the network needs to learn such that $\tilde{f}$ directly represents the count of the objects. In addition to computing $\tilde{f}$, we also computed the subtraction $\tilde{f}_{sub}$ of each pair of $\tilde{f}$. This was done by computing the product in the Laplace domain between each pair of $F$ as described in the previous section. The total number of units was 180 (20 units per module, 3 independent modules and 6 subtraction modules). When other RNNs were used, the dense layer was mapped to 180 recurrent units.

The output of the recurrent layer was passed to an actor network and a critic network. Both actor and critic consist of a single layer fully connected neural network. We set the discount rate to $\gamma = 0$ (since in this task, the reward was immediately available to agents after they made a correct turn). For all agents, we explored two different learning rates 0.001 and 0.0001.

# 3 RESULTS

## 3.1 PERFORMANCE OF RL AGENTS

We trained and evaluated agents in three different RL environments: accumulating towers, range count and exact count. We compared several agents based on the Laplace framework: the proposed agent using the inverse Laplace transform and the subtraction ($\tilde{f}_{sub}$), without the subtraction ($\tilde{f}$),

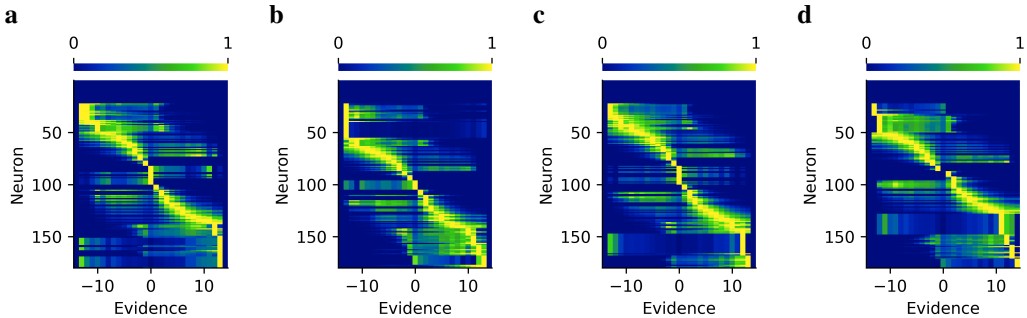

Figure 5: Neural activity of ($\tilde{f}_{sub}$) agents after 100k episodes of accumulating towers task. Similar to plots in Nieh et al. (2021); Morcos & Harvey (2016b), neurons are sorted by peak activity. Each row is normalized such that the activity ranges from 0 to 1.

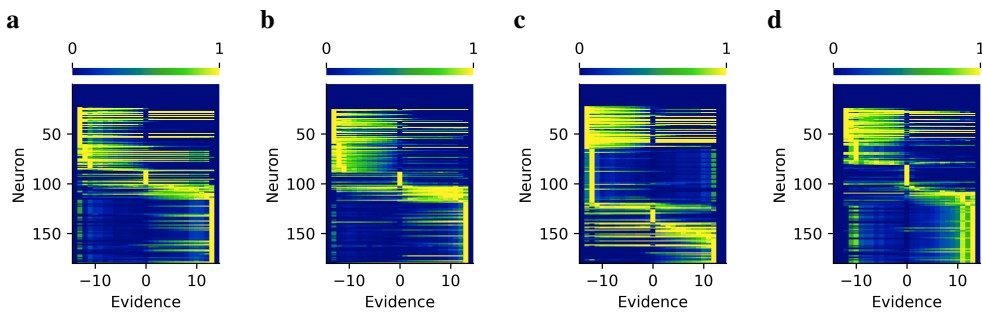

Figure 6: Same as Fig. 5 but for GRU agents.

with the subtraction but without the inverse Laplace transform ($F_{sub}$), and without both subtraction and the inverse Laplace transform ($F$). This was done to evaluate the importance of different components of the Laplace framework. We also compared agents based on existing RNNs, including a simple RNN and GRU, as well as versions of those agents but with frozen recurrent weights. The control experiment with frozen recurrent weights was conducted because the Laplace agents have analytically computed recurrent weights $s$ which are modulated on a population level with $\alpha$ as shared weight.

| | d=300 | d=3000 | d=10000 | # Parameters |
|---|---|---|---|---|
| $\tilde{f}_{sub}$ | **10.000 ± 0.000** | **10.000 ± 0.000** | **10.000 ± 0.000** | 724 |
| $\tilde{f}$ | 9.903 ± 0.066 | 9.925 ± 0.065 | 9.925 ± 0.041 | **244** |
| $F_{sub}$ | **10.000 ± 0.000** | **10.000 ± 0.000** | **10.000 ± 0.000** | 724 |
| $F$ | 8.995 ± 0.287 | 8.675 ± 0.361 | 8.900 ± 0.272 | **244** |
| RNN | 4.475 ± 0.456 | 4.600 ± 0.281 | 4.700 ± 0.576 | 34024 |
| GRU | 9.98 ± 0.000 | 9.600 ± 0.146 | 8.425 ± 0.504 | 100624 |
| RNN$_{FROZEN}$ | −21.0 ± 0.000 | −201.0 ± 0.000 | −601.0 ± 0.000 | 724 |
| GRU$_{FROZEN}$ | −14.473 ± 5.653 | −149.3 ± 44.77 | −449.4 ± 131.3 | 724 |

Table 1: Results of agents trained of the accumulating towers task. The table shows mean reward +/- standard error across four runs after 100k episodes of training in d=300 steps long environment. Validation was done in 300, 3000 and 10000 steps long environments.

The agents were trained and evaluated in 300 steps long environment. We trained four different agents for each of the eight models and performed 100 validation runs every 1000 episodes.

| | d=300 | d=3000 | d=10000 | # Parameters |
|---|---|---|---|---|
| $\tilde{f}_{sub}$ | $\mathbf{10.000 \pm 0.000}$ | $\mathbf{10.000 \pm 0.000}$ | $\mathbf{10.000 \pm 0.000}$ | 724 |
| $\tilde{f}$ | $\mathbf{10.000 \pm 0.000}$ | $\mathbf{10.000 \pm 0.000}$ | $\mathbf{10.000 \pm 0.000}$ | $\mathbf{244}$ |
| $F_{sub}$ | $9.325 \pm 0.074$ | $9.525 \pm 0.096$ | $9.400 \pm 0.122$ | 724 |
| $F$ | $9.150 \pm 0.075$ | $9.350 \pm 0.115$ | $9.375 \pm 0.147$ | $\mathbf{244}$ |
| RNN | $-3.225 \pm 8.033$ | $-45.550 \pm 44.875$ | $-145.2 \pm 131.6$ | 34024 |
| GRU | $9.125 \pm 0.758$ | $4.15 \pm 0.557$ | $4.300 \pm 0.696$ | 100624 |
| $\text{RNN}_{FROZEN}$ | $-31.00 \pm 0.000$ | $-201.0 \pm 0.000$ | $-601.0 \pm 0.000$ | 724 |
| $\text{GRU}_{FROZEN}$ | $-31.00 \pm 0.000$ | $-201.0 \pm 0.000$ | $-601.0 \pm 0.000$ | 724 |

Table 2: Results of agents trained of the range count task. The table shows mean reward +/- standard error across four runs after 100k episodes of training in d=300 steps long environment. Validation was done in 300, 3000 and 10000 steps long environments.

| | d=300 | d=3000 | d=10000 | # Parameters |
|---|---|---|---|---|
| $\tilde{f}_{sub}$ | $\mathbf{10.000 \pm 0.000}$ | $\mathbf{10.000 \pm 0.000}$ | $\mathbf{10.000 \pm 0.000}$ | 724 |
| $\tilde{f}$ | $\mathbf{10.000 \pm 0.000}$ | $\mathbf{10.000 \pm 0.000}$ | $\mathbf{10.000 \pm 0.000}$ | $\mathbf{244}$ |
| $F_{sub}$ | $\mathbf{10.000 \pm 0.000}$ | $\mathbf{10.000 \pm 0.000}$ | $\mathbf{10.000 \pm 0.000}$ | 724 |
| $F$ | $7.975 \pm 0.129$ | $8.250 \pm 0.075$ | $8.275 \pm 0.246$ | $\mathbf{244}$ |
| RNN | $7.975 \pm 0.222$ | $8.000 \pm 0.146$ | $8.225 \pm 0.222$ | 34024 |
| GRU | $9.375 \pm 0.375$ | $8.050 \pm 0.545$ | $8.175 \pm 0.219$ | 100624 |
| $\text{RNN}_{FROZEN}$ | $-21.10 \pm 8.574$ | $-148.7 \pm 45.27$ | $-448.8 \pm 131.8$ | 724 |
| $\text{GRU}_{FROZEN}$ | $-21.23 \pm 8.465$ | $-148.7 \pm 45.27$ | $-448.8 \pm 131.8$ | 724 |

Table 3: Results of agents trained of the exact count task. The table shows mean reward +/- standard error across four runs after 100k episodes of training in d=300 steps long environment. Validation was done in 300, 3000 and 10000 steps long environments.

## 3.2 AGENTS BASED ON THE COGNITIVE MODEL LEARNED THE TASKS FASTER THAN GRU-BASED AGENTS

While in some cases other agents had a better start, $\tilde{f}_{sub}$ agents were the first to learn the tasks and converge to a reward value of 10 (see supplemental video of $\tilde{f}_{sub}$ agent performing the accumulating towers task). This indicates that the cognitive model provided a good representation, and once the A2C algorithm learned to use it, it was able to perform perfectly (Fig. 4 and the first column in Table 1, Table 2, and Table 3; see also Fig. 13, Fig. 14, and Fig. 15 for results on a wider training range). In accumulating towers task and exact count task, $F_{sub}$ agents converged next. While the difference between $\tilde{f}_{sub}$ and $F_{sub}$ agents is only in the inverse Laplace transform which is just a linear projection, this result suggests that the sequential map-like activation in $\tilde{f}_{sub}$ was more useful in learning to perform the task with perfect accuracy than the Laplace representation with exponentially decaying traces. $\tilde{f}$ and $F$ agents performed above chance (agents reached the end of the track and made a correct decision in more than 50% of cases), but did not reach the performance of 10 during 100k episodes on accumulating towers task suggesting that the subtraction operation which constructed a map-like representation for evidence was important for learning.

GRU agents managed to reach performance close to 10, indicating that they can learn the tasks as well (see supplemental video of GRU agent performing the accumulating towers task). On the other hand, the RNN agents did not learn the tasks in 100k episodes indicating that gating was important for correct performance. It is important to note that the cognitive model also constructed the representation of evidence by a gating mechanism. Similarly, GRU can learn to modulate the range of change in neural activity by the amount of change in the evidence. Frozen models were not able to learn the task, failing to even reach the end of the track and make a random decision as indicated by the total reward being negative.

### 3.3 AGENTS BASED ON THE COGNITIVE MODEL WERE ROBUST TO CHANGES IN THE ENVIRONMENT

To test the ability of agents to generalize, we also evaluated them on 3000 and 10000 steps long tracks without ever training them on tracks of that length (second and third column in Table 1, Table 2 and Table 3). Agents based on the cognitive model showed great resilience to this kind of rescaling. This is not surprising since the representation was designed to change as a function of change in the amount of evidence, so rescaling the environment did not have any impact. On the other hand, the performance of GRU agents dropped at unseen track lengths but remained well above chance. This suggests that GRU agents were able to learn to modulate their activity by the change in the amount of evidence: when there was no new evidence, there was little to no change in the activity, making the impact of track rescaling relatively small.

We also evaluated agents in environments that had 1 to 30 towers on each side instead of 1 to 14 towers (Table 4). While all agents suffered some loss in reward, GRU and $\tilde{f}_{sub}$ seem most resilient to this change, indicating potential similarity in the form of those two representations.

### 3.4 NEURAL ACTIVITY INSIDE THE RECURRENT LAYER RESEMBLES ACTIVITY IN MICE HIPPOCAMPUS

We visualized the neural activity of each of the four agents for each of 8 models after 100k episodes of training on the accumulating towers task. As expected, neurons in $\tilde{f}_{sub}$ agents activated sequentially as a function of evidence resembling the activity in neural recordings from the hippocampus (Nieh et al., 2021; Morcos & Harvey, 2016b) (Fig. 5). $\tilde{f}$ neurons also showed some tuning to the amount of evidence, but since they were able to only count objects on each side (and not subtract them), the tuning is blurry (Fig. 7). Neurons in $F$ and $F_{sub}$ agents showed gradual changes as a function of evidence rather than tuning to a particular magnitude of evidence reflecting the exponential dynamics of the Laplace transform (Fig. 8 and Fig. 9 respectively).

Some of the neurons in GRU agents showed tuning to the magnitude of evidence with often prominent asymmetry between positive and negative amount of evidence (Fig. 6). (See supplemental video of changes in the neural representation during training of GRU agent.) In comparison to neurons in RNN agents (Fig. 11), GRUs had a firing pattern significantly more dependent on the amount of evidence. Neurons in frozen GRU agents also showed some tuning to the magnitude of evidence (Fig. 10), although less than neurons in trained GRU agents.

## 4 CONCLUSIONS

We evaluated different artificial agents on three tasks 1) evidence accumulation task that mimicked the procedure of a recent neuroscience study (Nieh et al., 2021; Morcos & Harvey, 2016b), 2) range count task, and 3) exact count task. We used a simple setup with only three inputs and compared agents based on a cognitive model with agents based on simple RNN and GRU.

Agents based on a cognitive model were able to learn faster and generalize better despite having almost three orders of magnitude fewer parameters than GRU-based agent (244 vs 100624). This is an indicator that the A2C algorithm was able to use the neural representation from the cognitive model. This representation also resembled data from neural recordings in Nieh et al. (2021); Morcos & Harvey (2016b) characterized with the sequential activation as a function of the amount of evidence.

Agents based on the cognitive model represented perceptual information on a logarithmic scale, consistent with the Weber-Fechner law. This type of representation saves resources and can explain human behavioral outputs in perceptual tasks.

While we focused on experiments that involve numerosity, the proposed architecture can represent any latent variables (e.g., size, distance or luminosity) as a log-compressed number line. The only condition is that the latent variables change over time such that their time derivative can be extracted from the input signal (see Fig. 3 for examples of log-compressed number lines for discrete and continuous signals).

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

# A  APPENDIX

| | d=300 | d=3000 | d=10000 | # Parameters |
|---|---|---|---|---|
| $\tilde{f}_{sub}$ | $9.892 \pm 0.067$ | $\mathbf{9.850 \pm 0.056}$ | $\mathbf{9.925 \pm 0.041}$ | 724 |
| $\tilde{f}$ | $7.060 \pm 0.220$ | $7.175 \pm 0.343$ | $6.800 \pm 0.242$ | $\mathbf{244}$ |
| $F_{sub}$ | $8.845 \pm 0.151$ | $8.775 \pm 0.137$ | $8.525 \pm 0.248$ | 724 |
| $F$ | $7.293 \pm 0.257$ | $7.625 \pm 0.410$ | $7.350 \pm 0.109$ | $\mathbf{244}$ |
| RNN | $4.538 \pm 0.449$ | $4.800 \pm 0.429$ | $4.775 \pm 0.461$ | 34024 |
| GRU | $\mathbf{9.895 \pm 0.016}$ | $9.725 \pm 0.054$ | $9.025 \pm 0.219$ | 100624 |
| $\text{RNN}_{FROZEN}$ | $-21.000 \pm 0.000$ | $-201.0 \pm 0.000$ | $-601.0 \pm 0.000$ | 724 |
| $\text{GRU}_{FROZEN}$ | $-14.555 \pm 5.582$ | $-149.6 \pm 44.5$ | $-449.7 \pm 131.0$ | 724 |

Table 4: Mean reward +/- standard error across four runs on accumulating towers task after 100k episodes of training in d=300 steps long environment and 1 to 14 towers shown to agents. Validation was done in 300, 3000 and 10000 steps long environments and with 1 to 30 towers shown to agents.

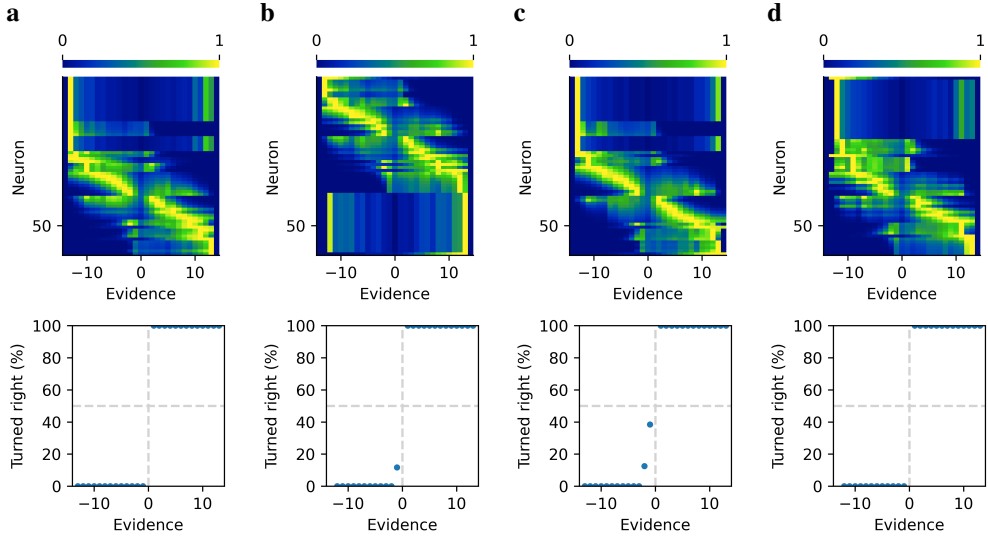

Figure 7: Neural activity (top row) and psychometric curves (bottom row) of $\tilde{f}$ agents after 100k episodes of accumulating towers task. Top row conveys the same information as Fig. 5. Since $\tilde{f}$ did not have perfect accuracy, we also show the psychometric curves. Dots in the psychometric curves indicate actions taken by the agent at different amounts of evidence.

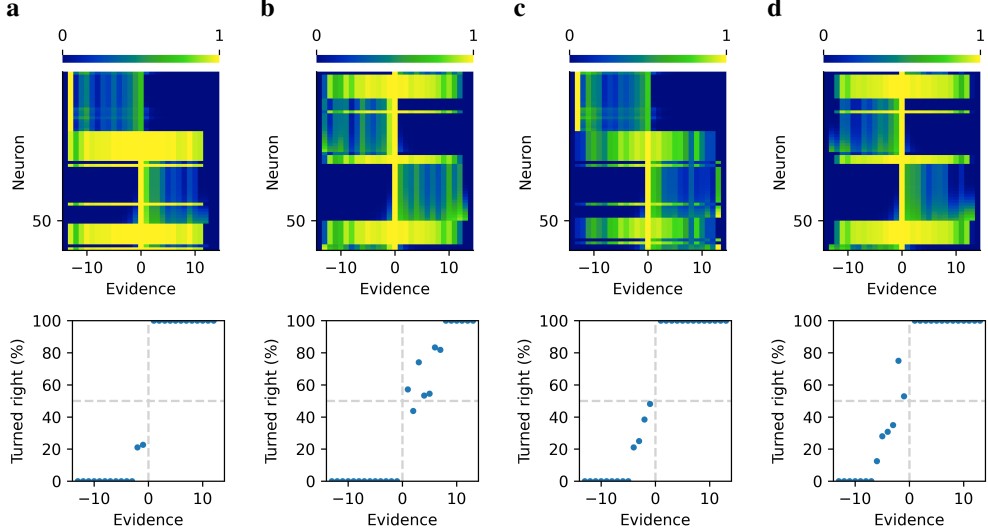

Figure 8: Same as Fig. 7 but for $F$ agents

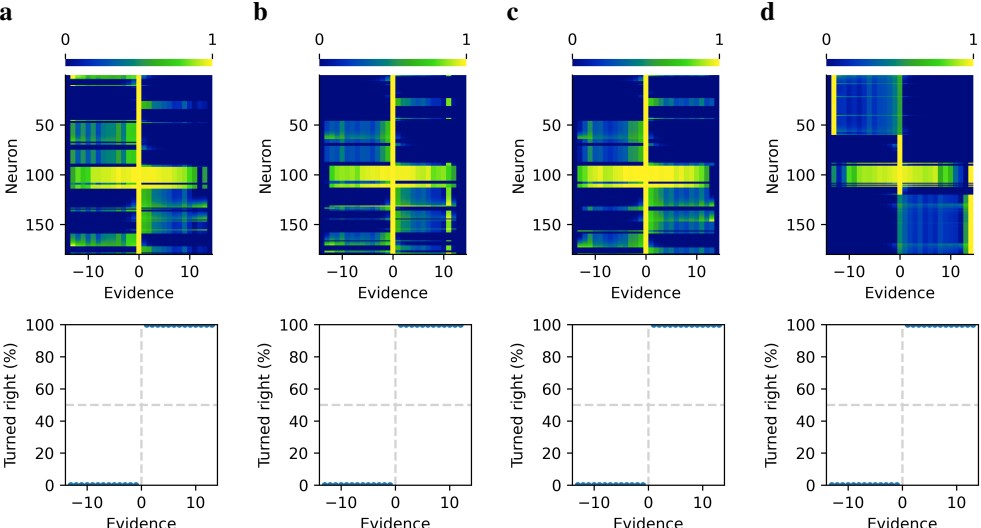

Figure 9: Same as Fig. 7 but for $F_{sub}$ agents.

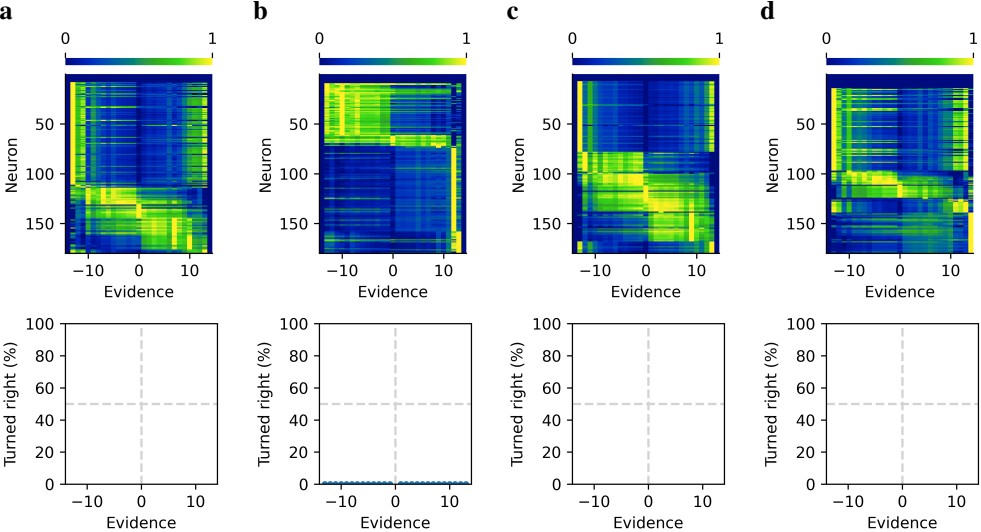

Figure 10: Same as Fig. 7 but for GRU$_{frozen}$ agents. Note that some psychometric curves are empty, indicating that those agents did not learn to reach the end of the environment.

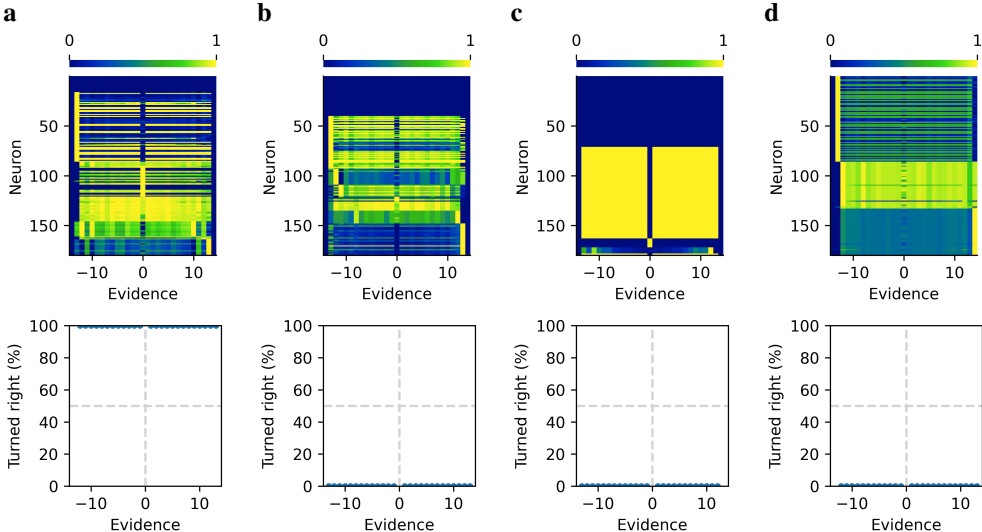

Figure 11: Same as Fig. 7 but for RNN agents.

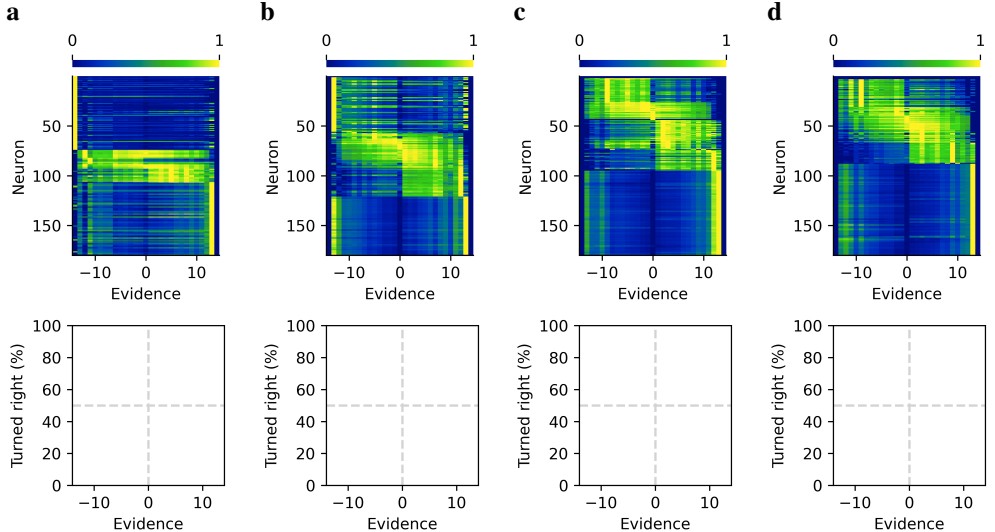

Figure 12: Same as Fig. 7 but for RNN$_{frozen}$ agents. Note that psychometric curves are empty, indicating that the agents did not learn to reach the end of the environment.

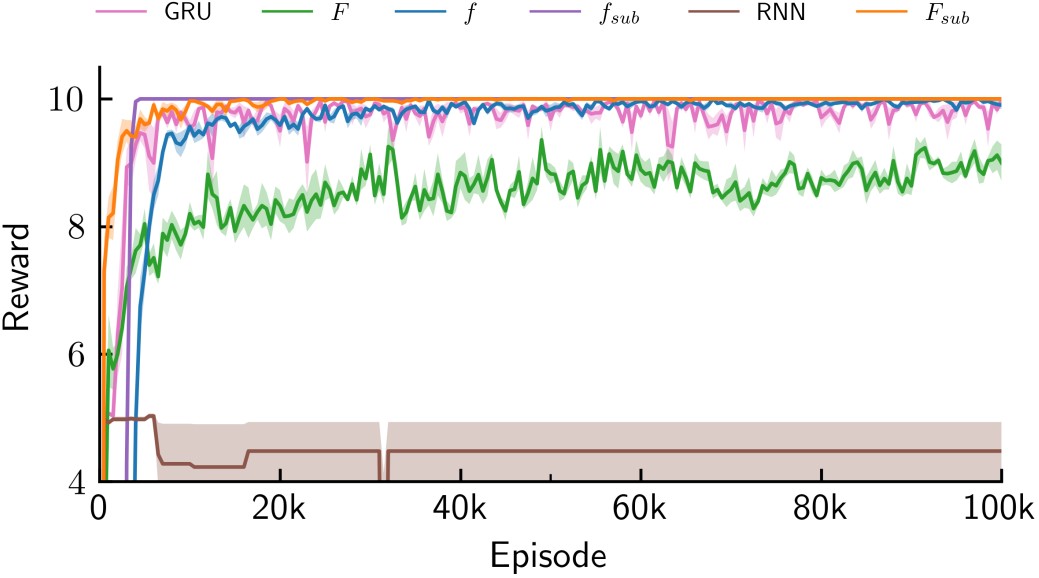

Figure 13: Agent performance on the accumulating towers task (same data as in Fig. 4a but different scale).

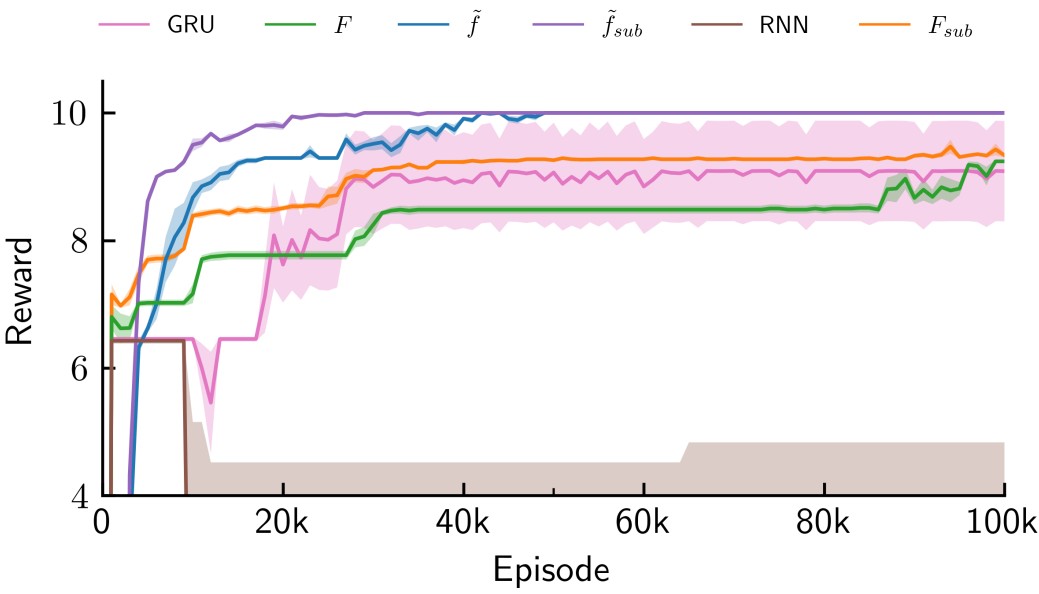

Figure 14: Agent performance on the range count task (same data as in Fig. 4b but different scale).

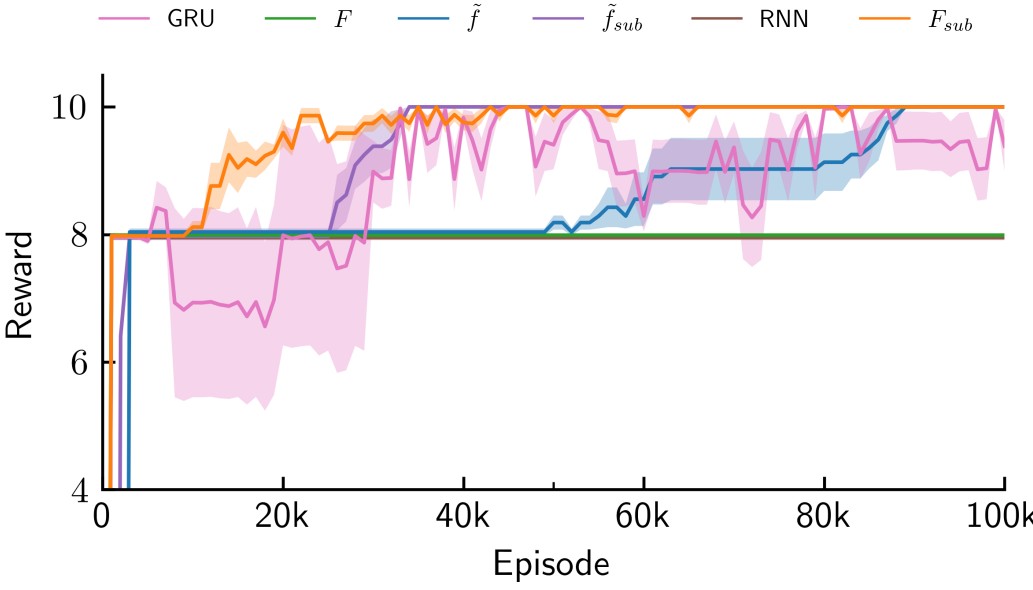

Figure 15: Agent performance on the exact count task (same data as in Fig. 4c but different scale).

