# OpenReview forum: "Characterizing neural representation of cognitively-inspired deep RL agents during an evidence accumulation task"
_ICLR.cc/2023/Conference — Submitted to ICLR 2023_

### Official Review · Reviewer_BAao · 2022-10-25

**Confidence:** 4
**Correctness:** 3
**Technical Novelty And Significance:** 3
**Empirical Novelty And Significance:** 2
**Recommendation:** 5

**Clarity, Quality, Novelty And Reproducibility:**

The current paper presents a novel idea of using Laplacian transformation to develop/inspire a recurrent network. However, the writing is not very clear (see my comments above) and some typo exists. It is unclear how technically the Laplacian transformation is embedded into the recurrent network. Since this is a small network model and I believe the paper can be reproduced without too much effort.

**Strength And Weaknesses:**

It seems to be novel to use Laplacian transformation to derive a neural representation of evidence or numerosity. I believe this framework may suggest some general principle of neural coding. Nevertheless, I feel some parts of the paper are not well written and can be improved further.

### Major concerns

- Based on the text it is unclear to me that how the authors embed the Laplacian transformation into a recurrent network, even if I read the manuscript several times. Do authors firstly perform a Laplacian transformation on the input, and then feed the transformed inputs to a recurrent net? Or do authors use the Laplacian transformation to propose or constrain the recurrent net structure?

- The connection between Eqs. 3 and 7 implicitly suggests that the firing rate of neurons represents the Laplacian transformation of an input. It is better to have more detailed explanations at here.

### Minor concerns

- If I understood correctly, Eq. 3 is wrong because it is inconsistent with Eq. 2, while Eq. 4 is correct. There must be some typos.

- It is not clear of the meaning of n* in Eq. 5. Is it the preferred evidence of a neuron? Also, is $\tilde{f}(n*, n)$ the reconstruction of $f(t)$ by only using a Laplacian basis?

- $\alpha(t_i)$ is not defined and I need to guess. Is it the instantaneous change of numerosity as mentioned right above Eq. 3? Besides, is the modulatory input $(\alpha=1)$ also the change of numerosity? If so, I don't understand why $\alpha=1$ represents no modulatory input.

- The definition of $W_L$ right below Eq. 7 is contradictory with each other and quite confusing.

- The math definition in Eq. 8 is no problem but I don't understand why author call the cross-correlation as subtraction between functions.

- The 4th line to the 7th line below Eq. 8 is repetitive.




**Summary Of The Paper:**

This paper embeds a learnable recurrent network based on Laplacian transformation into a reinforcement learning agent and claims that the Laplacian based recurrent network outperforms the others.

**Summary Of The Review:**

I have gone through the math derivations and read the manuscript for several times and I believe I try my best to understand the paper but there are still some places which are not clear.

---

> ### Author Response · Authors · 2022-11-19
> **Initial response to Reviewer's comments**
>
> The reviewer asked how the Laplace transform is embedded into a neural network. In the methods section, we showed that the real-domain Laplace transform can be approximated as a recurrent neural network with a diagonal connectivity matrix. In other words, a set of neurons with exponentially decaying impulse responses that span a wide range of (log-spaced) time constants. We clarified these statements in the revised version of the manuscript, as well as the connection between Eqs. 3 and 7, which, as the reviewer pointed out, indeed implies that a recurrent neural network can implement the Laplace transform.
>
> The reviewer was concerned that Eq. 3 might be wrong. This is not the case since Eq. 3 is an extension of Eq. 2 - we clarified the text to avoid any ambiguity. Eq. 2 is a simple differential form of the standard Laplace transform, while Eq. 3 expands that form by adding shared modulation of the recurrent weights.
>
> The reviewer asked about variable $n^*$ in Eq. 5. As the reviewer stated, it is indeed the preferred evidence of a neuron (it is the amount of evidence for which the activity of a neuron indexed with $n^*$ peaks). Also, $F(n^*,n)$ is the reconstruction of $f(n)$ using only the Laplace basis.
>
> $\alpha(t)$ is defined as a time derivative of numerosity ($\alpha = dn/dt$) in the line above Eq. 3 and in a more general form as a time derivative of any latent variable ($\alpha = dx/dt$) in the top paragraph on page 5.
>
> We agree with the reviewer that the definition of $W_L$ was confusing, and we made changes to the methods section on page 5 to make it easier to follow (also, $W_L$ is now just labeled as $W$).
>
> The reviewer asked why we call the cross-correlation a subtraction between functions. While this manuscript is not the first to state this property, it is, in general, not widely known. To gain intuition about it, imagine taking a cross-correlation between two delta functions, e.g., $\delta(a_1)$ and $\delta(a_2)$. The result of cross correlation will be a function $\delta(a_2-a_1)$. This intuition can be expanded to general functional forms, not just delta functions, and gives rise to the subtraction of fuzzy numbers. Similarly, convolution between two functions gives rise to addition (this is why the 4th and 7th line below equation 8 do not mean the same thing: one describes addition, and the other describes subtraction - we added a clarification there to avoid confusion). Convolution and cross-correlation can be easily implemented in the Laplace domain as elementwise products. We believe this is an important contribution of the proposed work since by building the Laplace transform as a part of a differentiable neural network, we are opening doors for using different computations that can be implemented in the Laplace domain (note that subtraction of fuzzy numbers outside the Laplace domain is a non-trivial problem without an obvious neural network implementation).
>
> **In general, we see this work as a step towards integrating symbolic computations (such as subtraction) that are critical for human cognition into neural networks.** While in our main experiment agents had to learn to subtract the number of towers from each side of the environment, the potential applications of this approach are much broader since navigation in any physical or abstract space commonly involves subtraction, for example, computing a spatial distance between two locations, time between two events or difference in ferociousness between two animal species.

---

### Official Review · Reviewer_ojuW · 2022-10-26

**Confidence:** 3
**Correctness:** 3
**Technical Novelty And Significance:** 1
**Empirical Novelty And Significance:** 2
**Recommendation:** 5

**Clarity, Quality, Novelty And Reproducibility:**

The paper is clear and very easy to follow. I also appreciate its reproducibility particularly because of the provided supplementary material. The modelling/theoretical contribution as well as experiments/results however is limited.

**Strength And Weaknesses:**

Strength: The paper looks into an interesting and commonly used task in decision making and RL experiments in the field of Neuroscience.
The paper is clearly written and different components are very well explained step by step. It also provides a virtual environment, useful for the community to simulate evidence accumulation experiments.

Weakness: This work only covers one specific task from one family of experiments (Evidence accumulation). The paper can be significantly improved if the method is tested on broader range of tasks in the face of uncertainty (e.g. see "Recurrent Model-Free RL Can Be a Strong Baseline for Many POMDPs" by Ni et al, ICML 2022), and/or works on hippocampus and cognitive maps (many references  presented in the paper) and/or different evidence accumulation tasks in neuroscience (e.g. see works of Michael Shadlen lab for human and monkey, and Carlos Brody lab for rodent). In terms of architecture, comparison of an architecture with a Bayes filter (e.g. see implementations of "End-to-end learnable histogram filters", by Bonschkowski & Brock, NeurIPS deep RL workshop 2016, or "QMDP-Net: Deep Learning for Planning under Partial Observability" by Karkus et al, NeurIPS 2017) instead of Laplace framework could be illuminating.


**Summary Of The Paper:**

The paper describes a network that combines a Laplace framework into a reinforcement learning architecture achieve superior performance over GRU and RNN in an evidence accumulation task both in terms of reward gain and generalization. Moreover, the activity of units in the proposed architecture resembles map-like neural activity in the brain for accumulated evidence.

**Summary Of The Review:**

I think because of lack of significant modeling/theory contribution as well lack of test-domains this paper is not ready to be presented in the ICLR venue yet.

Update after rebuttal:
I appreciate the authors' response and adding more experiments. The added experiments, however, are very much in the same domain of previous one and only a threshold has been changed (the accumulation mechanism is the same). In the evidence accumulation task (e.g. in mentioned labs), a sufficiently task has either 1) different number of options (e.g. more than 2) 2) different decision time criterion (reaction time v.s. fixed) or 3) dynamic weigh of evidence through the task. Moreover, even if the proposed approach is new, we do not know whether it is works better than a simple Bayesian update, which is in fact shown to be able to solve evidence accumulation task.

---

> ### Author Response · Authors · 2022-11-19
> **Initial response to Reviewer's comments**
>
> The reviewer mentioned that “This work only covers one specific task from one family of experiments (Evidence accumulation)” and suggested several alternatives. To address this valid concern, we designed two more experiments inspired by the work of the authors mentioned by the reviewer (Shadlen and Brody). In the first new task, the agents had to count a specific number of objects, and in the second one, the agents had to identify a target range of a number of objects. As we mentioned in response to Reviewer 1, the task of counting objects is analogous to well-known behavioral tasks such as interval timing (e.g., if objects were present on the screen the entire time, counting the number of objects is equivalent to measuring time) and estimating spatial distance (where the flow of objects corresponds to the optic flow). In each of these tasks, the proposed architecture showed faster learning and better generalization than GRU-based agents which failed to reach perfect performance after 100k training episodes.
>
> The reviewer also expressed concern about the “lack of significant modeling/theory contribution“. This is also an important concern, and we improved the original manuscript by stressing the theoretical significance of the methods section. Adding the capability of modulating the recurrent weights by some function of the input enables the neural activity to change gradually as a function of latent variables. **The proposed approach here differs fundamentally from existing gated RNNs since the gated weights are shared across a population of neurons.** We showed that this is equivalent to implementing the Laplace transform of those latent variables. **To the best of our knowledge, this is a novel concept with important theoretical significance and promising utility.** The inverse of the transform then gives a structured representation where a set of neurons activates sequentially as a function of the magnitude of the latent variable. Our results indicate that this structured representation facilitates learning and generalization.
>
> As we also mentioned in response to Reviewer 1, we added a discussion on page 9 and emphasized that the concept of weight modulation with shared weights can represent any latent variables that change over time. The reason we focused on the accumulating towers task is the existence of neural data that show activity remarkably similar to the activity generated by the equations. In general, if some feature (e.g., size, distance, luminosity or numerosity) changes as a function of time, then the proposed network contains computational machinery to discover those latent variables from the input signal and represent them as a log-compressed number line.

---

> ### Comment · Area_Chair_xvBj · 2022-12-08
> **Do the new experiments address your concerns?**
>
> Dear reviewer,
>
> I'm curious if the new experiments the authors included address your concerns about the limited nature of their tasks? Please take a look at their response as well as updated paper with the new experiments.
>
> Thank you for your time,
> Area chair

---

### Official Review · Reviewer_tPG3 · 2022-11-03

**Confidence:** 3
**Correctness:** 4
**Technical Novelty And Significance:** 3
**Empirical Novelty And Significance:** 2
**Recommendation:** 6

**Clarity, Quality, Novelty And Reproducibility:**

**Clarity**
- Section 2 is quite clear given the math in it; while it did take a couple reads to build intuition for the Laplace framework, it wasn't overly daunting or impossible
- Results section could use better claims-based signposting, rather than topic-based, or just a general info dump as most of it is now (which is not to say that long section doesn't read well; it does). Three of the four contributions are claims that will be demonstrated, so the results section may benefit from section headings based on those claims.

**Quality**
- Experiment section is carefully considered. The experiments designed to support each claim in the contributions are intuitive and convincingly designed
- The paper doesn't leave any loose ends or unanswered questions. It feels like a finished piece of work.

**Originality**
- The high-level concept of cognitive models as inductive biases (or some such integration) is familiar, but I believe this particular approach for evidence accumulation is novel. I think the contribution list is quite an accurate and complete summary of the parts of this paper that are novel.

Nits:
- "Amount of evidence" is strange phrasing here. It seems from the rest of the paper and the place cell example that this means *n* itself, but the phrase sounds like it means, how much evidence we have that the value is *n* as opposed to other values.

**Strength And Weaknesses:**

**Strengths**
- Introduction is strong and motivates the paper well. The idea of cognitive models providing inductive biases is not novel but this is a very interesting instantiation of it
- The connections to human cognitive maps are more convincing than a lot of "human cognitive construct + our cool new method" connections that ML papers attempt to make, especially because of follow-up evidence in the paper such as the logarithmic representation. While even this one is ultimately conceptual (and the logarithmic representation appears to be a design choice rather than an emergent property, if I understand correctly), I find it compelling.
- The paper is well-packaged - it makes several claims and demonstrates them, and the methods solve exactly the problem motivated and outlined. To some degree, the final contribution about gating could use more expansion, but I realize it's more a validation of prior work and I appreciate the acknowledgement of the similarities/power of existing work

**Weaknesses**
- The Laplace framework presented here seems very tuned to this specific task, or at least counting-based tasks. The paper acknowledges in "Future Work" that this is by no means the only type of latent variable and encourages work on others, but as of right now the framework is engineered in terms of counting - or at least, that's the sense I get reading it. It's still an interesting method, but GRUs and RNNs are much more general modules; it would be helpful to have (brief!) discussion of either 1) the domain limitations of these equations or 2) the potential to easily adapt this method to other tasks. I see how this formulation may extend to some other quantities, but I don't know for sure.
- The discussion I described above would make the comparisons in this paper make more sense, but for actual acceptance I'm not sure that would be enough. This is an interesting idea and novel to my knowledge, but without seeing experiments in other domains, it's not clear how much value it adds. A more general approach covering a larger domain (potentially through multiple frameworks like this one), and experimental results on those, would be more convincing.
- From a pure utility perspective, it's unclear how much value this method provides over GRU (not to mention, it has a narrower domain if I understand correctly). Because the performances are so similar, some statistics differentiating them would help.
- The word "deep" is used in the title and throughout the paper, but the version of A2C experimented on uses single-layer FCNs for actor and critic. The paper isn't making claims about scale or ability to handle large real-world tasks or complicated environments, so this claim seems not only false but unnecessary to the story.

**Summary Of The Paper:**

The paper presents a novel framework for reinforcement learning, specifically for evidence accumulation tasks that involve counting quantities and deciding which is greater. The general approach is to use cognitive models as inductive biases by learning a function representing the evidence and giving the resulting representations to an RL algorithm. The paper includes a discretized version of the Laplace transform-based framework to use with a neural network.

The framework is as follows: the Laplace transform is rewritten as a differential equation in terms of the numerosity of the stimulus and discretized; an inverse is also derived. This is meant to enable learning of a counted quantity. To subtract one quantity from another and determine which one is higher, the paper presents a cross-correlation mechanism.

This framework integrates into an RL algorithm (here A2C) by having environment inputs go into a recurrent layer, have that layer learn a direct representation of the count of the objects and their difference (in theory), and then pass the output of this layer to the actor and critic networks.

Experiments are done on an "accumulating towers" task. There are several ablations removing various parts of the Laplace framework, as well as RNN and GRU layers as baselines. Task reward statistics show that the Laplace and inverse Laplace transforms with subtraction do very well, as does GRU. Other ablations have mid-range performance, and RNN does quite poorly. Frozen versions of RNN and GRU are unsurprisingly very poor.

When evaluating on harder versions of the task, Laplace with subtraction continues to do well, as does GRU. The paper attributes this to the gating mechanisms present in both methods.

Finally, there is some analysis of neural activity and its connections to human cognitive maps. Results show that Laplace with subtraction appears to activate sequentially as a function of evidence. GRU and other versions of the Laplace framework also shows some tuning to "magnitude"/"amount of evidence".

**Summary Of The Review:**

I am recommending a weak acceptance for this paper. It is a good paper and I enjoyed reading it, but I am not convinced of its significance from a general modeling perspective given the limitations I think I see in its domain. I also am not totally sure I understood all the math, so I may be missing limitations; I may also be missing strengths.

All that said, the quality of idea and execution is far above a workshop paper and it is a very compelling read, so I wouldn't consider it fair to recommend rejection just because it's an interesting and well-demonstrated idea rather than the next big SOTA system like a lot of papers. I would raise my score if I were convinced that this opens a lot of future work directly (not just in high-level concept) or enables improvement on new tasks out of the box, and if I were convinced that it provides meaningful improvement over existing methods.

---

> ### Author Response · Authors · 2022-11-19
> **Initial response to Reviewer's comments (part 1)**
>
> The reviewer mentioned that “the logarithmic representation appears to be a design choice rather than an emergent property”. Interestingly, scale-invariance, which is necessary for logarithmic compression, is actually an emergent property of the inverse Laplace transform computed using Post inversion (as done in this manuscript). This is manifested through a linear increase in the coefficient of variation of the bell-shaped tuning curves as a function of the peak time (see paragraph below Eq. 6 in revision). This linear relationship implies that on a log-scale bell-shaped tuning curves are equally wide. To ensure they are also equidistant, we chose log-spaced time constants of units that compute the Laplace transform. Logarithmic representation is widely present in the human perceptual system. It is implemented through physical receptors (e.g., in the retina and cochlea). However, log compression is apparent from behavioral experiments that include time, numerosity, or spatial distance (well-established through Weber’s law), which do not have physical receptors. Therefore the Laplace transform can be a mechanism for building log compression in neural systems. **The proposed approach equips artificial neural networks with this human-like representational property. This aligns with the primary objective of our work, which is the development of AI systems that compute in a human-like manner.**
>
> The reviewer suggested expanding more on the importance of gating, which is listed as one of the contributions of our manuscript. RNNs, by default, change gradually as a function of time. Adding the capability of modulating the recurrent weights by some function of the input enables networks to change gradually as a function of latent variables whose derivatives can be extracted from the input. **Here is where the proposed approach differs fundamentally from existing gated RNNs since the gated weights are shared across a population of neurons. We showed that this is equivalent to implementing the Laplace transform of those latent variables** (Eq. 4). The inverse of the transform then gives a structured representation where a set of neurons activates sequentially as a function of the magnitude of the latent variable. **Our results indicate that this structured representation facilitates learning and generalization.**

---

> > ### Author Response · Authors · 2022-11-19
> > **Initial response to Reviewer's comments (part 2)**
> >
> > The reviewer raised a concern that “the Laplace framework presented here seems very tuned to this specific task or at least counting-based tasks.” and suggested “adding a brief discussion of either 1) the domain limitations of these equations or 2) the potential to easily adapt this method to other tasks.” We added a discussion on page 9 (last paragraph) and emphasized that the concept of weight modulation with shared weights can represent any latent variables that change over time. The reason we focused on the accumulating towers task is the existence of neural data that show activity remarkably similar to the activity generated by the equations. In general, if some feature (e.g., size, distance, luminosity or numerosity) changes as a function of time, then the proposed network contains computational machinery to discover those latent variables from the input signal and represent them as a log-compressed number line. We also added Fig. 3c which illustrates that this approach can represent continuous latent variables.
> >
> > To further address the generality of the proposed approach, we designed two more experiments, one in which the agents had to count a specific number of objects and another one in which agents had to identify a target range of a number of objects. In each of these tasks, the proposed architecture showed faster learning and better generalization than GRU-based agents which failed to reach perfect performance after 100k training episodes. Note that while the task domain remained the same, the task of counting objects is analogous to well-known behavioral tasks such as interval timing (e.g., if objects were present on the screen the entire time, counting the number of objects is equivalent to measuring time) and estimating spatial distance (where the flow of objects corresponds to the optic flow).
> >
> > Following a suggestion from the reviewer, we omitted the word “deep” when discussing our agents since the neural network we used effectively had one trainable layer.
> >
> > We also reorganized the results section following a suggestion from the reviewer to include subheadings that can guide readers through the results and better reflect the main contributions.
> >
> > The reviewer asked about differences in performance as compared to GRU-based agents. We emphasize that the proposed agents were, in general, faster at learning and better at generalization (e.g., performing better at unseen track lengths) while having almost three orders of magnitude fewer parameters than GRU (244 parameters vs. over 100 thousand). This radical saving in the number of parameters comes primarily from the concept of shared recurrent weights.

---

### Author Response · Authors · 2022-11-19
**Summary of changes in the revised manuscript**

We thank the reviewers for their time and deeply appreciate the constructive comments they made. In response to their suggestions, we conducted two additional experiments and added the results to the revised version of the manuscript. Following suggestions from the reviewers, we also rewrote parts of the manuscript to emphasize the theoretical contributions of this work. We hope that these new changes will justify the importance of this work.

Here we emphasize a few key points related to the revision, and in the comments below, we provide detailed responses to comments from each reviewer:

1. The proposed approach implements a cognitive model as a differentiable artificial neural network giving rise to a representation that resembles neural activity in the brain, and that can facilitate learning as demonstrated in out experiments.

2. The main theoretical contribution of this work is in showing that the cognitive model, which is based on the Laplace transform, can be implemented through the gating of recurrent connections with shared weights. Weight sharing in RNNs is a novel concept that gives rise to a structured representation that has properties of a number line (much like CNNs give rise to a structured output due to shared weights and dense layers do not).

3. As proof of concept, we demonstrate our approach on the “accumulating towers task”, which strongly impacted the neuroscience community (the study that inspired our main experiment was published in Nature in 2021). We showed that the neural representation that emerged in the neuroscience study can result from a neural network gated with shared weights.

4. The proposed approach is not limited to the “accumulating towers task” but can be used to discover different latent features that change over time (e.g., size, distance, or luminosity). To illustrate the generality of our approach, in the revision, we added two more experiments (results are in Fig. 4b and 4c and Tables 2 and 3) where the proposed approach overperforms agents based on GRUs. We also added a figure that shows how our approach can be used for discovering continuous latent variables such as distance (Fig. 3c).

---

> ### Author Response · Authors · 2022-12-11
> **Author follow up**
>
> Dear reviewers, as the discussion period is coming to an end, we wanted to thank you again for your detailed and helpful feedback. We hope our responses adequately addressed your concerns. Please let us know if you have any outstanding questions and we will reply promptly.

---

### Decision · Program_Chairs · 2023-01-20

**Decision:**

Reject

**Justification For Why Not Higher Score:**

The reviews for this paper were a bit mixed, with some noting that the experiments were too limited, while another found the links to neuroscience to be compelling and the overall work above that of a workshop paper. On balance, I didn't find the reasons for acceptance to outweigh the weaknesses noted in the reviews.

**Justification For Why Not Lower Score:**

N/A

**Metareview: Summary, Strengths And Weaknesses:**

This paper presents a model for evidence accumulation using a Laplace transform-based framework embedded in a recurrent RL architecture trained in an “accumulating towers” task, inspired by animal neuroscience. They find that, compared to vanilla GRU and RNN-based RL agents, their approach achieves better performance and generalization, and that there is evidence of the emergence of brain-like representations, although this evidence seems mixed.

Initially, the reviewers didn’t particularly feel strongly one way or another, with 2 recommending weak reject and 1 recommending weak accept. They indicated that they liked the motivation of the work and the ties to neuroscience which are interesting, and found it overall well-presented. However there were concerns that experiments were only performed on a single task, limiting the generality of their conclusions.

In the rebuttal period, the authors added experiments on two additional tasks, but reviewers noted that since they were in the same domain and were not meaningfully different from the original task, this was not enough to satisfactorily address this main criticism. Reviewer ojuW also indicated that the work was missing comparison to additional models from the decision-making literature and is thus still not ready for publication.

During subsequent discussions, all reviewers continued to express concerns that this paper wasn’t quite ready for publication, including tPG3, who was previously most positive.

In the end, reviewers appeared to be in agreement that this paper would benefit from more time, a wider range of tasks, and more model comparisons. Therefore, I cannot recommend acceptance, but look forward to future iterations of this work with reviewers’ feedback incorporated.